# Effects of Developmental Timing on Cognitive and Behavioral Profiles in Fetal Alcohol Spectrum Disorder: Considerations for Education

**DOI:** 10.3390/bs14060431

**Published:** 2024-05-22

**Authors:** Nerea Felgueras, José María López-Díaz, Inmaculada Garrote

**Affiliations:** High-Performance Research Group on Inclusive Education, People with Disabilities and Universal Accessibility (DIVERSIA), Department of Educational Sciences, Faculty of Education and Sport and Interdisciplinary Studies, King Juan Carlos University, 28933 Fuenlabrada, Spain; inmaculada.garrote@urjc.es

**Keywords:** childhood, adolescence, executive functions, behavioral problems, prenatal alcohol exposure, educational needs

## Abstract

Associations and families demand the need to raise awareness of the implications in the cognitive and behavioral development of children with Fetal Alcohol Spectrum Disorder (FASD) that affect their learning and school participation. This study aims to generate a profile of executive and behavioral functioning in children and adolescents diagnosed with FASD. A probabilistic sampling by clusters (associations for individuals with FASD) is applied. The sample is composed of 66 families from three associations. The BRIEF-2 and SENA tests were administered to assess executive and behavioral functioning domains. Data analysis found that the executive and behavioral functioning profile of individuals with FASD varies with age, with greater impairment in middle and late adolescence. Likewise, the domain of executive functioning most affected in any of the developmental stages is working memory. Finally, cognitive impairment in the executive functioning domains has a direct impact on the social and adaptive functioning of people with FASD.

## 1. Introduction

Fetal Alcohol Spectrum Disorder (FASD) brings together a heterogeneity of conditions related to prenatal alcohol exposure associated with facial abnormalities; unusual growth and weight (considering established age parameters); and irregularities in brain structures and neurocognitive difficulties, including impaired intellectual capacity, learning difficulties, memory impairment, and deficits in visuospatial ability, executive functioning, and self-regulatory capacity. It is considered the leading non-genetic cause of completely preventable intellectual disability [1].

The literature surrounding this disorder calls for an accurate and universal detection and diagnostic system that enables early identification of its core manifestations [2,3]. Based on these arguments and considering the prevalence data in Spain, estimated at 22.2% cases of FASD per 10,000 inhabitants [4], as well as the lack of knowledge among healthcare and educational professionals regarding this disorder [5,6,7,8,9,10,11], it is essential to have a behavioral and cognitive profile that considers the developmental stage.

Therefore, research on FASD needs to progress towards evaluating the cognitive and behavioral domains affected by prenatal alcohol exposure, considering not only the clinical perspective but also the viewpoint of caregivers of individuals affected by this disorder [12]. Available evidence indicates that families and teachers are the best informants of the observable behaviors of children and adolescents with this diagnosis [13,14].

Regarding executive functions, these are defined as a multidimensional construct referring to interrelated abilities that enable a person to direct their behavior towards a goal [15]. The most representative impairments in executive functions in individuals with FASD are difficulties in organizational and planning abilities, concrete thinking, inhibition problems, understanding cause–effect relationships, following instructions, developing a plan of action with a predefined objective, making informed judgments, generalizing, and understanding abstract concepts [2]. At the same time, behavioral impairments are understood as a set of maladaptive behaviors depending on the specific circumstances, where the common element lies in difficulties with behavior, self-control, and emotion regulation [16]. The most probable behavioral impairments in FASD include difficulties with emotion regulation, behavioral manifestations of executive function dysregulation, attention deficit, hyperactivity, impulsivity, irritability, sleep disorders, deficits in social skills, and difficulties in adaptive behavior and social communication [2]. Likewise, the influence of gender on the expression of difficulties in executive and behavioral functioning in cases of comorbidity with ADHD should be considered [17]. In this sense, gender plays an important role in the profile of cognitive and behavioral functioning, with a greater deterioration in males [18].

The most significant implications of behavioral disorders are the transgression of people’s rights, as well as of established social norms, which often leads to repeated problems with justice and serious difficulties in adapting to the surrounding environment. Social ignorance about the implications of the behavioral alterations of this disorder in the autonomous life of the affected persons favors limitations in participation in society [19,20,21].

This study aims to create a profile of executive and behavioral functioning of the people with FASD according to developmental stage, gender, and diagnosis through the contributions of the family member who has the greatest opportunities for observation of the affected person.

The following working hypotheses are considered:It is observed that the contribution of families through the BRIEF-2 and SENA assessments allows for the development of a cognitive and behavioral functioning pattern for children and adolescents with FASD.It is contemplated that age is a factor that modulates significant differences in the neurocognitive and behavioral profiles of individuals affected by FASD.

## 2. Materials and Methods

### 2.1. Study Design

The Research Ethics Committee of the Universidad Rey Juan Carlos issued a favorable opinion in September 2020 on the adequacy of the planning and research design of this study, meeting the necessary ethical requirements in relation to the objectives of the research project. Furthermore, all participants in this study agreed to the requirements of this study and consented to participate in this study.

It is a quasi-experimental design, being the only viable alternative in the impossibility of conducting a random assignment research, which could hinder the control of all variables involved in families’ responses. Additionally, it is identified with a quantitative methodology. It is characterized by its objectivity and the use of tools that allow the precise measurement of specific psychological constructs, enabling the statistical treatment of the data and the generalization of the results to the population.

The method followed in the study is shown in the flow chart below (Figure 1).

### 2.2. Participants

The attainment of the primary research unit was carried out following a probabilistic cluster sampling approach, resulting in five conglomerate groups. Each group corresponds to one of the registered associations nationwide related to FASD and Fetal Alcohol Syndrome (FAS), regulated under the provisions of the Organic Law 1/2002, dated 22 March, regulating the Right of Association. The sample size for each of the conglomerates is distributed as follows: 250 cases registered in VISUAL TEAF (stands for FASD), 305 cases in AFASAF (stands for Association of Families Affected by Fetal Alcohol Syndrome), 80 cases in SAFGROUP (stands for FAS), 60 cases in Tolerancia-Cero, and 40 cases in Zero FAS (stands for FAS). Subsequently, three groups were selected according to random sampling rules: VISUAL TEAF, SAFGROUP, and AFASAF. The sample consists of a total of 66 families of children with FASD. Regarding the association of origin, 40.9% belong to AFASAF (n = 27), 27.3% to SAFGROUP (n = 18), and 31.8% (n = 21) are associated with VISUAL TEAF. The distribution of participants includes 57.6% men (n = 38) and 42.4% women (n = 28). The breakdown by age groups is as follows: 19.7% (n = 13) in the child group, 25.8% (n = 17) in the early adolescence group, 21.2% (n = 14) in the middle adolescence group, and 33.3% (n = 22) in the late adolescence group. There are 49 cases with a complete Fetal Alcohol Syndrome (FAS) diagnosis (n = 49; 74.24%), 11 cases of partial FAS (n = 11; 16.66%), and 6 cases of Alcohol-Related Neurodevelopmental Disorder (ARND) (n = 6; 9.09%). If attention is paid to the co-morbidity situation, there are six cases presenting a joint diagnosis with Attention Deficit and/or Hyperactivity Disorder (n = 6; 9.09%); four cases showing Specific Learning Disorder (n = 4; 6. 06%); four cases showing Conduct Disorder (n = 4; 6.06%); two cases presenting hypoacusis (n = 2; 3.03%); a single case presenting Specific Language Disorder (n = 1; 1.51%); and, finally, two cases presenting comorbidity with Autism Spectrum Disorder (n = 2; 3.03%). The rest of the co-participants did not present comorbidity with other disorders (n = 47; 71.21%). The diagnosis of all cases was made based on the criteria established by Hoyme [1]. Finally, 78.8% had a disability (n = 52), 48.5% were in a situation of dependency, and, lastly, all participants were adopted (n = 66; 100%).

### 2.3. Variables

On the one hand, the developmental stage or age, gender, and the diagnosis of FASD are considered independent variables. The developmental stage is organized into childhood (7–10 years), early adolescence (11–14 years), middle adolescence (15–17 years), and late adolescence (18–19 years). The gender variable distinguishes between men and women. The variable diagnosis of FASD includes complete FAS, partial FAS, and ARND.

On the other hand, the dependent variables are identified with each of the executive functioning domains of the BRIEF-2 scale: inhibition (INH), self-monitoring (SMO), flexibility (FLE), emotional control (EMC), initiative (INI), working memory (WM), planning and organization (PLA), task monitoring (TAS), and materials organization (ORG). Finally, the dependent variables are also associated with each of the domains assessed in the SENA test: depression (DEP), anxiety (ANS), social anxiety (SCA), somatic complaints (SOM), attention problems (ATE), hyperactivity–impulsivity (HIP), anger control problems (ANG), aggression (AGG), challenging behavior (CHA), antisocial behavior (ANT), substance use (SUB), eating behavior problems (EAT), unusual behavior (UNU), emotional regulation problems (REG), rigidity (RIG), isolation (ISO), social integration and competence (SOC), emotional intelligence (EMI), and study disposition (STU).

### 2.4. Measurement

In this study, the standardized test Behavior Rating Inventory of Executive Function, Second Edition, hereinafter BRIEF-2, in its Spanish adaptation, was applied. In addition, the standardized neuropsychological test ‘Sistema de Evaluación de Niños y Adolescentes’ (SENA) is applied for the Spanish-speaking population. This test is applied because of its close parallelism with the diagnostic criteria included in the DSM-V [16] and its adequate psychometric properties. The administration of both tests uses a self-administered version aimed at families of children and adolescents with FASD.

The BRIEF-2 and SENA assessments are structured according to three age levels, which primarily correspond to the educational stages of Early Childhood Education, from 3 to 6 years old; Primary Education, from 6 to 12 years old; and finally, Compulsory Secondary Education, from 12 to 18 years old. However, they allow for some flexibility to adapt to the specific assessment needs based on the stage of the developmental cycle.

The administration of the Behavior Rating Inventory of Executive Function, Second Edition (BRIEF-2) provides a profile of impairment in different areas of executive function in children and adolescents aged 5 to 18 years, with an approximate administration time of 10 min. The BRIEF-2 consists of 63 items that are assessed on a Likert-type rating scale with three response options: never, sometimes, and frequently. Additionally, the System for Evaluating Children and Adolescents (SENA) is also administered. It evaluates various psychological problems frequently encountered in children and adolescents from a multidimensional perspective. The SENA family version for adolescents contains 154 items, using a Likert-type rating scale with five response options, where respondents are asked to select the frequency of a behavior. The SENA family version for the child population (6 to 12 years old) includes 129 items, utilizing the same rating scale. Both versions have an approximate administration time of 30 min. One aspect to highlight is that there is consistent evidence of the validity and reliability of the results provided by both tests. In addition, there is correlation between these two tests in studies of populations with neurodevelopmental disorders [13,14].

## 3. Statistical Methods and Results

To create the executive and behavioral functioning profile, the typical scores (T) obtained from the BRIEF-2 and SENA tests will be used for each variable. Additionally, data analysis is performed using the SPSS V27 software for Windows.

The normality of the independent variables (developmental stage and diagnosis) and the dependent variables of executive and behavioral functioning are assessed using the Kolmogorov–Smirnov test (D) for the total sample (n ≥ 50) and the Shapiro–Wilk test (W) for groups with n ≤ 50. Regarding the sociodemographic variable of the developmental stage, it is determined that all subgroups of this variable do not follow the assumption of normality, as the significance level is ≤0.05 (D = 0.170, p_total_ < 0.000; W = 0.821, p_childhood_ < 0.012; W = 0.874, p_early-adolescence_ < 0.025; W = 0.862, p_middle-adolescence_ < 0.032; W = 0.628, p_late-adolescence_ < 0.000). The diagnostic variable does not meet the assumption of normality in its distribution, with *p* < 0.000 (D = 0.448). Table 1 shows the normality study for the dependent variables.

To ascertain if there are significant differences in the variables of executive and behavioral functioning between the general population and the clinical population (FASD), Student’s *t*-test for independent samples is applied in cases where the assumption of normality is met. It is important to note that the general population mean is 50, with a standard deviation of 10, in both the BRIEF-2 test [14] and the SENA test [13]. Student’s test of executive and behavioral functioning is presented in Table 2.

If *p* < 0.05, the data are inconsistent with the hypothesis that the population mean value is as proposed. If the lower and upper bounds include the value zero, the sample data are compatible with the proposed population value, so the H_0_ is accepted, and vice versa.

Bivariate correlation analysis is used to determine whether there are differences according to age and diagnosis for each of the variables. It will allow us to find out whether there is an association between the variable age or diagnosis and each of the variables, as well as the strength and directionality of this correlation (bilateral probability) (Table 3 and Table 4). The program automatically calculates the level of significance from the 99% and 95% confidence interval, so that significant correlations at the 0.01 level will be identified with two asterisks (**) and significant correlations at the 0.05 level with one (*).

It is noted that age is a factor that intervenes in almost all the variables of executive control and behavioral functioning.

In addition, the diagnosis does not determine the level of executive functioning. However, it does correlate with the behavioral functioning variables attention problems, hyperactivity, unusual behavior, integration problems, and social competence and willingness to study.

This analysis is not sufficient to explain the nature of this correlation, so one-factor analysis of variance (ANOVA) is applied. It determines whether the population means of the independent variable (IV) and the dependent variable (DV) are equal. This test assumes the principle of homoscedasticity, or equality of variances, and the normality of distribution. The higher the F-statistic, the more the variables are related, i.e., DV values differ between VI groups. If the *p* is less than 0.05, the null hypothesis (H_0_) of equal means is rejected, concluding that not all compared population means are equal (Table 5 and Table 6).

In turn, the Kruskal–Wallis H-test is applied to determine whether there are significant differences in the executive and behavioral functioning variables according to the type of diagnosis (Table 7 and Table 8). If the critical level is less than 0.05, the null hypothesis of equality of population means is rejected and it can be concluded that diagnostic types differ on the cognitive and behavioral functioning variables.

Therefore, in the sample assessed there are no statistically significant differences in cognitive and behavioral functioning according to diagnosis.

Finally, differences in gender in the variable of executive and behavioral functioning will be tested by applying the non-parametric Kolmogorov–Smirnov test for independent samples (n ≤ 30) (Table 9).

In the executive functioning variables, no statistically significant differences were observed between men and women. However, this is not true for the variables working memory (*p* < 0.001) and task monitoring (*p* < 0.013), as they have a significance level of less than 0.05. On this occasion, the null hypothesis is retained in which the behavioral functioning variables belong to the same population, regardless of gender, except for the variables ATE (attention problems (*p* < 0.005)) and ISO (isolation (*p* < 0.011)), which reflect significant differences according to male or female sex.

## 4. Discussion

In the first instance, the BRIEF-2 neuropsychological test provides a measure of executive function assessment based on an ecological perspective [14]. There are three previous studies that apply the BRIEF-2 test for the assessment of all executive functions in the population affected by FASD in the United Kingdom and Canada, showing significant alterations in most of the domains assessed by the test, except in the domain “organization of materials” [12,18,22]. The results of this study follow a similar pattern to those obtained in different works [12,18,23]. That is, participants affected by FASD show significant differences in the executive functioning domains assessed (inhibition, self-monitoring, flexibility, emotional control, initiative, working memory, planning and organization, task monitoring, and organization of materials), with working memory being the most impaired variable and organization of materials the least impaired, although in some cases they continue to belong to the clinical population. However, we find that the interaction of the age variable is a moderating factor for executive functioning [23]. Thus, school-aged children, which in this study comprises ages seven to ten, show less marked impairment in the executive functions of inhibition, self-monitoring, cognitive flexibility, emotional control, working memory, planning and organization, and task monitoring compared to their peers in higher developmental stages (middle and late adolescence). The results of this study coincide with those obtained in the studies by [23], in which age is identified as a factor that determines the deterioration of executive functioning, which is more notable at higher ages, but which stabilizes in the adolescent period. This not only confirms the hypothesis that the age variable is a determinant of the level of functioning of the different domains of executive functions at a general level, but also achieves one of the objectives of this study: the verification of significant differences in EF motivated by the evolutionary moment of the person affected by FASD. Finally, studies by several authors ([19,24,25], among others) analyze the comorbidity between FASD and ADHD and reveal that cases with a comorbid diagnosis with ADHD compared to a single diagnosis of FASD show higher scores in all domains of executive functions assessed by the BRIEF test, with the variables working memory and inhibition being the most affected. Thus, the presence of comorbid ADHD is associated with an exacerbation of impaired executive functioning [18,26].

Secondly, in the literature reviewed, there is no evidence of the application of the SENA test in the population affected by FASD. The impairment of neurocognitive domains affected by FASD, namely intellectual functioning, memory, attention, executive functions, or communication skills, directly impacts the person’s social and adaptive functioning. In other words, impaired executive functioning domains self-regulation, information processing, inhibition, working memory, self-monitoring, and planning and organization impact the ability to make decisions and regulate behavior effectively and adaptively [27]. Thus, executive functioning will condition behavioral functioning. Therefore, individuals exposed to alcohol in the prenatal period with impaired inhibition and emotional control experience serious behavioral problems, such as illogical thinking, difficulties in understanding cause–effect relationships, aggressive and antisocial behavior, or other maladaptive behaviors, irrespective of IQ. In this sense, most of the child-age population with severe adaptive behavioral disturbances also show them in adulthood, often associated with problems with justice [28]. This study shows the presence of alterations in all of the domains of adaptive behavior assessed by means of the SENA test in the clinical population affected by FASD in comparison with the general population. Specifically, alterations coinciding with the clinical population were observed in the domains of depression, anxiety, social anxiety, attention problems, hyperactivity and impulsivity, anger control problems, aggression, defiant behavior, antisocial behavior, substance use, problems with eating behavior, unusual behavior, rigidity, isolation, integration and social competence, emotional intelligence, and willingness to study. As can be seen, the impairment of behavioral functioning in the diagnosis of FASD is compatible with the impairments in the domains, as can be similarly reflected in the studies of several authors ([1,2,19,20,21,22,27], among others). Depression poses severe life-threatening risks for adolescents affected by FASD, such as suicidal ideation, self-harming behaviors, or suicide attempts [29]. Deficits in behavioral control are explained by deficits in behavioral self-regulation, behavioral inhibition, and adaptive functioning [21]. Problems with anger control, aggression, or defiant behavior towards authority figures are common in the FASD population, especially with advancing age [21]. These adaptive dysfunctions, together with behavioral inhibition and impulsive behaviors, are consistent with impaired social functioning, intensifying at older ages [20]. All these findings are confirmed by the results obtained in this study. Furthermore, all these impairments in adaptive behavior and social competence lead to difficulties in community participation. In addition, social deterioration with advancing age is also motivated by alterations in the social and communication domain [19], revealing a halt in the progression of social competence. Thus, since social and communication demands increase in adolescence and adulthood, social deterioration is indeed confirmed as age increases [21].

The presence or absence of dysmorphic traits with prenatal alcohol exposure does not necessarily imply significant differences in executive functioning [19,30]. These results are supported by those obtained in the same study. Therefore, considering that the association between neurocognitive impairment and dysmorphic traits is not statistically significant, it is suggested that the diagnosis of FASD should prioritize central nervous system deficits [31]. Other studies show differences in terms of which behavioral disturbances manifested according to the diagnostic entity [32,33], with the genetic component, the environment, and early adversity experiences becoming conditioning elements [21].

From another point of view, considering the gender variable as a moderating factor in the impairment of executive and behavioral functioning, in this study, no significant differences were observed in either the executive functioning or behavioral functioning variables as a function of gender. These results are confirmed by those obtained in the work of Herman and colleagues [17] or Rai and colleagues [18], which reveal no statistically significant differences between boys and girls on measures of impaired executive functioning using the BRIEF-2 test. Likewise, these results are consistent with those obtained in the study by Sakano and colleagues [21], where possible differences in behavioral domains could be due to other factors unrelated to FASD.

## 5. Conclusions

Firstly, it is considered that the contribution of the family through the BRIEF-2 and SENA standardized assessment tests for the Spanish-speaking population allows for the elaboration of a defined executive and behavioral profile of the child and adolescent with a diagnosis of Fetal Alcohol Spectrum Disorder, thus fulfilling the first working hypothesis of this study.

The domains of executive functioning that are affected in each of the profiles that are formed from the three classifying factors can be found in Appendix A, expressed through the qualitative indicators proposed by the BRIEF-2 test: typical scores between 0–59 are qualified as “no clinical significance”, T scores between 60–64 as “mild elevation”, T scores between 65–69 as “potentially clinical elevations” and, finally, T scores equal to or above 70 as “clinically significant elevations”. In contrast, it was determined that both the gender variable and the diagnostic entity did not become conditioning factors for the level of cognitive functioning or the level of executive function. Ultimately, the pattern of executive functioning becomes very similar between the diagnostic entities considered in this study, which allows us to confirm that the hypothesis about affinity between patterns of functioning of the diagnostic entities is fulfilled.

It is also noted that the behavioral functioning of people with FASD is significantly altered in comparison with their age and gender counterparts because of alterations in executive functioning. This altered behavioral functioning mainly responds to problems of self-regulation, evidencing problems in anger control or emotional regulation, hyperactivity and impulsivity, problems in maintaining attention, cognitive rigidity, and alterations in social interaction and competence related to unusual behavior, which on many occasions leads to situations of marked social isolation. In addition, anxiety, depression, defiant behavior, antisocial behavior, and aggressiveness may be observed. In all cases, the willingness to study has been affected, remaining stable at higher ages. This fact highlights the absence or insufficient support to offer experiences of academic success to people affected by FASD within the educational system. In turn, the effect on behavioral functioning worsens with increasing age in some domains of behavioral functioning: anxiety, attention problems, challenging behavior, antisocial behavior, unusual behavior, problems in emotional regulation, cognitive rigidity, and social isolation. This may be due to a multifactorial origin, such as the person’s support network, experiences of academic success, or perceived social support, among others (See Appendix B).

The scope of this research study in the field of research is promising. The development of a neurocognitive and behavioral profile of the adopted person with a diagnosis of FASD has shown that developmental timing is an element that modulates the expression of alterations in executive and behavioral functioning. However, this finding is subject to a more complex assessment that includes not only the analysis of the affected person’s environment, but also its comparison with a clinical group of the biological population. In any case, having a defined profile in terms of cognitive and behavioral functioning is a very valuable resource for professionals in the educational community. It serves on the one hand, as a guide to understanding how these students function and learn and, on the other hand, as a guideline for the implementation of inclusive educational practices. The main methodological limitations identified during the development of the research are the sample size, the lack of research, and the biases associated with the administration of questionnaires. Considering that there are no real data describing the prevalence and incidence data of this disorder in Spain, rather than the mere fact of an estimate of it [4], it is difficult to define a representative sample size of the clinical population. Although it is true that, considering the estimated data (22 people affected per 10,000 inhabitants), the sample size is small (n = 66). In addition, the establishment of the correlation between the alterations in executive and behavioral functioning and the age variable has been carried out based on the available sample. However, the ideal would be to design a longitudinal study to analyze the development of executive functions and the establishment of a behavioral pattern in the same population, which would undoubtedly require an increase in economic resources and time. In this sense, the conclusions drawn are limited to the conditions of this study. Finally, the bias associated with the administration of questionnaires that use the Likert scale as a recording measure can lead to the loss of information, since it is limited to polarized categories and can even be affected by the statistical treatment carried out by the researcher himself [34].

As lines of future research, it is recommended to continue with the design of specific training on FASD that can be offered in educational centers, through associations, in congresses or seminars, where the profiles generated in this study can be provided.

## Figures and Tables

**Figure 1 behavsci-14-00431-f001:**
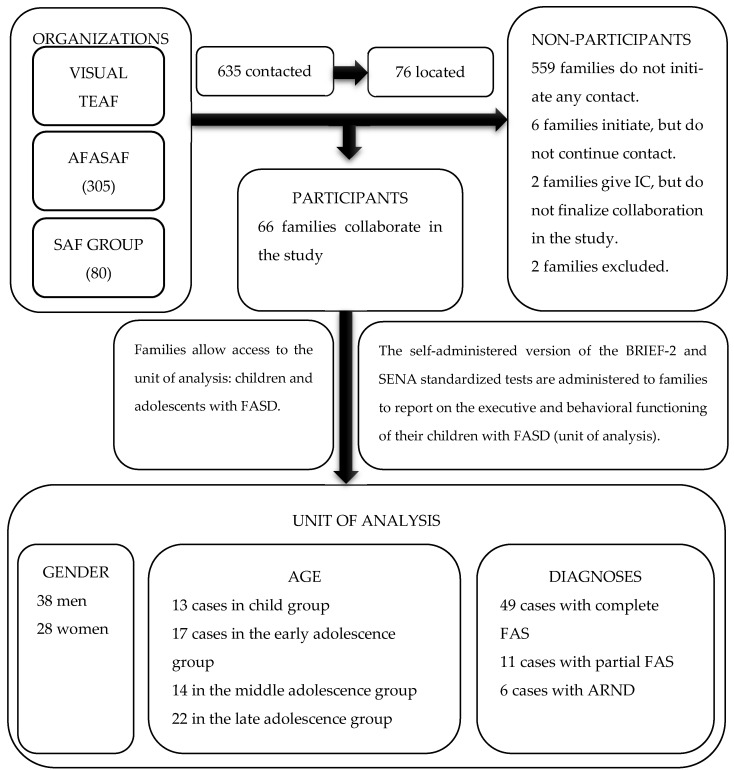
Research Method Flowchart.

**Table 1 behavsci-14-00431-t001:** Normality test.

Variable EF	FE Statistics		Variable BF	BF Statistics	
	D	*p*		D	*p*
INH	0.076	0.435	DEP	0.810	0.331
SMO	0.156	0.000	ANS	0.089	0.212
FLE	0.118	0.021	SCA	0.103	0.075
EMC	0.121	0.014	SOM	0.161	0.000
INI	0.094	0.150	ATE	0.079	0.376
WM	0.073	0.503	HIP	0.102	0.079
PLA	0.181	0.000	ANG	0.092	0.170
TAS	0.116	0.026	AGG	0.123	0.013
ORG	0.094	0.151	CHA	0.113	0.032
			ANT	0.191	0.000
			SUB	0.362	0.000
			EAT	0.166	0.001
			UNU	0.103	0.077
			REG	0.124	0.011
			RIG	0.110	0.043
			ISO	0.106	0.062
			SOC	0.133	0.005
			EMI	0.091	0.191
			STU	0.135	0.004

N = 66. D = Kolmogorov-Smirnov statistic with Lilliefors significance correction (n ≥ 50) based on 10,000 Monte Carlo samples with a starting seed of 2,000,000. If *p* ≥ 0.05 H_0_ is accepted so the observed values are distributed under the assumption of normality.

**Table 2 behavsci-14-00431-t002:** Student’s *t*-test of EF and BF domains.

Variable EF	*t*-Test Results EF		Variable BF	*t*-Test Results BF	
	T	*p*		T	*p*
INH **	15.22	0.000	DEP **	8.38	0.000
SMO *	17.71	0.000	ANS **	11.37	0.000
FLE *	17.52	0.000	SCA **	5.79	0.000
EMC *	11.86	0.000	SOM *	2.02	0.047
INI **	23.33	0.000	ATE **	20.22	0.000
WM **	22.38	0.000	HIP **	13.04	0.000
PLA *	26.81	0.000	ANG **	10.7	0.000
TAS *	20.13	0.000	AGG *	8.45	0.000
ORG **	12.87	0.000	CHA *	10.6	0.000
			ANT *	9.32	0.000
			SUB *	3.46	0.001
			EAT **	3.72	0.001
			UNU **	11.83	0.000
			REG *	14.05	0.000
			RIG *	16.77	0.000
			ISO **	13.66	0.000
			SOC *	−15.37	0.000
			EMI **	−13.79	0.000
			STU *	−19.36	0.000

** normal distribution; * non-normal distribution; N = 66; df = 65; test value = 50. *p* = bilateral significance.

**Table 3 behavsci-14-00431-t003:** Analysis of bivariate correlations in EF and BF variables according to age.

Variable EF	Pearson	Spearman’s Rho	p_EF_	Variable BF	Pearson	Spearman’s Rho	p_BF_
INH	0.456 **		0.000 **	DEP	0.105		0.401
SMO		0.529 **	0.000 **	ANS	0.274 *		0.026 *
FLE		0.487 **	0.000 **	SCA	0.019		0.877
EMC		0.473 **	0.000 **	SOM		−0.015	0.907
INI	0.153		0.219	ATE	0.290 *		0.018 *
WM	0.275 *		0.026 *	HIP	0.251 *		0.042 *
PLA		0.335 **	0.006 **	ANG	0.263 *		0.033 *
TAS		0.427 **	0.000 **	AGG		0.379 **	0.002 **
ORG	0.163		0.183	CHA		0.352 **	0.004 **
				ANT		0.566 **	0.000 **
				SUB		0.258	0.070
				EAT		0.100	0.491
				UNU	0.340 **		0.005 **
				REG		0.434 **	0.000 **
				RIG		410 **	0.001 **
				ISO	0.263 *		0.033 *
				SOC		0.004	0.976
				EMI	−243 *		0.049 *
				STU		−0.152	0.222

** correlation is significant at the 0.01 level; * correlation is significant at the 0.05 level. N = 66. *p* = bilateral significance. If *p* ≥ 0.05, H_0_ is accepted, i.e., there are no differences in the executive/behavioral functioning variable as a function of age.

**Table 4 behavsci-14-00431-t004:** Analysis of bivariate correlations in EF and BF variables according to diagnosis.

Variable EF	Pearson	Spearman’s Rho	p_EF_	Variable BF	Pearson	Spearman’s Rho	p_BF_
INH	−0.034		0.786	DEP	0.021		0.434
SMO		0.030	0.813	ANS	−0.144		0.124
FLE		−0.085	0.497	SCA	0.045		0.359
EMC		0.044	0.726	SOM		−0.043	0.366
INI	−0.122		0.329	ATE	−0.257 *		0.019
WM	−0.090		0.471	HIP		−0.262 *	0.017
PLA		0.054	0.668	ANG	0.228		0.033
TAS		0.001	0.996	AGG		0.097	0.220
ORG	−0.240		0.052	CHA		−0.010	0.468
				ANT		−0.075	0.275
				SUB		0.110	0.190
				EAT		−0.062	0.312
				UNU	−0.264 *		0.016
				REG		0.108	0.195
				RIG		−0.188	0.066
				ISO		0.025	0.422
				SOC	−0.293 **		0.008
				EMI	−0.085		0.249
				STU		0.215 *	0.041

** correlation is significant at the 0.01 level; * correlation is significant at the 0.05 level. N = 66. *p* = bilateral significance. If *p* ≥ 0.05, H_0_ is accepted, i.e., there are no differences in the executive/behavioral functioning variable as a function of diagnosis.

**Table 5 behavsci-14-00431-t005:** One-factor ANOVA for the age variable and the EF and BF variables.

ANOVA EF			ANOVA BF		
EF	F	*p*	F	*p*	BF
INI	5.703	0.002	ANS	7.845	0.000
SMO	9.274	0.000	ATE	3.511	0.020
FLE	7.518	0.000	HIP	2.278	0.088
EMC	3.907	0.013	ANG	1.039	0.381
WM	4.430	0.007	AGG	2.610	0.059
PLA	3.534	0.020	CHA	2.911	0.041
TAS	6.075	0.001	ANT	5.695	0.002
			UNU	3.955	0.012
			REG	6.693	0.001
			RIG	6.434	0.001
			SOC	3.045	0.035
			EMI	0.954	0.420

Inter-group df = 3; intra-group df = 62; total df = 65.

**Table 6 behavsci-14-00431-t006:** One-factor ANOVA for the diagnosis variable and the EF and BF variables.

ANOVA BF		
F	*p*	BF
ATE	2.232	0.116
HIP	2.450	0.094
UNU	2.509	0.089
SOC	.322	0.726
STU	2.365	0.102

Inter-group df = 2; intra-group df = 63; total df = 65.

**Table 7 behavsci-14-00431-t007:** Kruskal–Wallis H-test of executive functioning variables according to diagnosis.

Variables EF	Average Range			Z	*p*
	FAS	pFAS	ARND		
INH	33.97	34.41	28	0.548	0.760
SMO	33.28	32.23	37.67	0.341	0.843
FLE	34.43	31.36	29.83	0.474	0.789
EMC	33.18	31.27	40.17	0.889	0.641
INI	35.14	25.73	34.33	2.186	0.335
WM	34.07	35.86	24.50	1.536	0.464
PLA	33.20	29	44.17	2.504	0.286
TAS	33	43.45	19.33	6.310	0.043
ORG	36.01	29.86	19.67	4.373	0.112

N_FAS_ = 49; N_pFAS_ = 11; N_ARND_ = 6; df = 2. Significance level is 0.05.

**Table 8 behavsci-14-00431-t008:** Kruskal–Wallis H-test of behavioral functioning variables according to diagnosis.

Variables BF	Average Range			Z	*p*
	FAS	pFAS	ARND		
DEP	32.99	37.77	29.83	0.801	0.670
ANS	35.41	27.73	28.50	1.894	0.388
SCA	32.92	35.27	35.00	0.176	0.916
SOM	33.97	32.41	31.67	0.120	0.942
ATE	36.30	29.41	18.17	5.388	0.068
HIP	36.51	24.00	26.33	4.750	0.093
ANG	31.70	31.50	51.83	6.031	0.049
AGG	32.71	30.45	45.50	2.713	0.258
CHA	33.78	29.91	37.83	0.702	0.704
ANT	34.30	31.95	29.83	0.384	0.825
SUB	32.32	36.14	38.33	0.795	0.672
EAT	34.39	27.55	37.17	1.411	0.494
UNU	36.45	26.36	22.50	4.674	0.097
REG	32.65	29.55	47.67	3.845	0.146
RIG	35.71	25.45	30.17	2.792	0.248
ISO	35.81	31.50	18.33	4.581	0.101
SOC	33.20	34.64	33.83	0.052	0.974
EMI	34.71	28.82	32.17	0.884	0.643
STU	31.32	35.41	47.83	4.105	0.128

N_FAS_ = 49; N_pFAS_ = 11; N_ARND_ = 6; df = 2. Significance level is 0.05.

**Table 9 behavsci-14-00431-t009:** Kolmogorov–Smirnov test for executive and behavioral functioning variables (grouping variable: gender).

Variables EF	Z	*p*	Variables BF	Z	*p*
INH	1.155	0.139	DEP	1.079	0.194
SMO	0.936	0.345	ANS	0.906	0.385
FLE	0.936	0.345	SCA	1.260	0.083
EMC	0.762	0.607	SOM	1.268	0.080
INI	0.694	0.721	ATE	1.743	0.005
WM	1.977	0.001	HIP	1.004	0.266
PLA	0.770	0.594	ANG	0.687	0.733
TAS	1.592	0.013	AGG	0.611	0.849
ORG	0.868	0.438	CHA	1.162	0.134
			ANT	0.574	0.897
			SUB	1.253	0.087
			EAT	1.004	0.266
			UNU	0.732	0.657
			REG	0.634	0.816
			RIG	1.230	0.097
			ISO	1.608	0.011
			SOC	0.845	0.473
			EMI	0.981	0.291
			STU	1.208	0.108

N_total_ = 66; N_men_ = 38; N_women_ = 28. Significance level is 0.05.

## Data Availability

The data presented in this study are available upon request to the corresponding author due to the preservation of the health data protection of the study participants.

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
