# Peer review of "Effects of Developmental Timing on Cognitive and Behavioral Profiles in Fetal Alcohol Spectrum Disorder: Considerations for Education"

_behavsci, 2024, doi:10.3390/bs14060431_

Round 1

Reviewer 1 Report

Comments and Suggestions for Authors

In general, the article presented to readers is interesting, but there are several things that need to be noted and need to be improved:

1. There is a need to improve the state of the art, especially discussing various emotional problems in students

2. The method described should use a flow chart

3. Explain what problems can be caused by FASD apart from affecting school activities

4. What is the difference between BRIEF-2 and Likert scale?

5. Has instrument validation been carried out on BRIEF-2?

Author Response

Por favor, consulte el archivo adjunto.

Reviewer 2 Report

Comments and Suggestions for Authors

Dear authors,

Thanks for the paper that is submitted, which covers an interesting topic. I think this article attempts to create a profile of executive and behavioural functioning in children and adolescents diagnosed with FASD. The article is well written and flows easily. I have major comments and edits that I would like to suggest to the authors:

The authors filiation is incomplete. It has to be updated.

In the Methods section related to the participants, it should be interesting to include how FASD is diagnosed.

In the Methods section related to measurement, the way of administration of the BRIEF-2 /SENA should be included: self or hetero administrated.

The ethics information related to Ethics Committee and informed consent should be included in the Method section.

I think a description of the bias that questionnaires involve should be included at the end of the discussion, as is described in this article:

Westland JC. Information loss and bias in likert survey responses. PLoS One. 2022;17(7):e0271949. Published 2022 Jul 28. doi:10.1371/journal.pone.0271949

Also, the limitations of the study should be included in the paper, such as sample size.

Line 328: FASD 

References:

Please, correct the references as the ACS style requires 

Reviewer 3 Report

Comments and Suggestions for Authors

The study is very interesting and addresses the topic of cognitive functioning of FASD people on which studies of this type are important to develop treatment strategies appropriate for age groups and developmental tasks.

The innovative aspect is the involvement of families in the study, a topic on which it would be necessary to focus attention both for treatment and prevention strategies.

The study captures one of the essential aspects of knowledge about FASD and confirms the need for treatment not only for FASD people but also for their families. I would suggest that the authors include bibliographical references on gender differences in the introduction, links between the aspects of cognitive functioning covered, prevention strategies and the involvement of families and school as prevention settings.

In the proposed methodology I appreciated the description of the participants and the tools. However, can the authors also specify the composition of the sample by gender? Can the authors specify whether the chosen tools are specific for FASD people? What guided their choice of these tools? Have they validated them for this specific population? Can the authors explain what the quantitative methodology consists of?

The conclusions are well articulated and consistent with the postulated working hypotheses; however, links to possible treatments and prevention strategies are lacking. Could the authors elaborate on what “future challenges” this study opens up to?

The bibliography is well articulated and updated and the tables are exhaustive. Is it possible for the authors to insert a table on socio-demographic variables and gender differences?

Paragraph 2.1 The paragraph of the study design is not very clear and in particular the meaning of "Additionally, it is identified with a quantitative methodology" is not clear. Could the authors explain the meaning of this phrase better?

Paragraph 2.3 I would suggest paying attention to the form, for example "on the other hand" is repeated in two close periods

Line 112 the authors mentioned the tools using the acronyms that are explained in lines 117 and 122; I would suggest writing the instruments in full on line 112 and using the acronyms in the following lines

Comments on the Quality of English Language

the manuscript is well written, I would suggest further reading to avoid the repetition of some terms ("on other hand" is often repeated)

Reviewer 4 Report

Comments and Suggestions for Authors

Dear authors, thank you for the oppotunity to review your article which focusses on executive and behavioral functioning in chidlren and adolescence with FASD. A well known problem in individuals with FASD with huge impact on for instance adaptive functioning in daily live. You were using a sample of 66 families from 3 associations with different ages and FASD diagnosis. I have a few things to mention. 

My questions / remarks to you: 

1. in Materials and Methods you are describing the participants and i must confess that i struggle a little bit with the different groups  VISUAL TEAF, SAFGROUP, AFASAF you are using. I am not sure what all these abbreviations mean and where thes paricipants come from. 

2. whos diagnosed the them and which diagnostic system was used (I assume Hoyme). 

3. Most og the participants have even FAS or pFAS, why did you include only a few with milder forms

4. you say that all participnats were adopted? Thats unusual compared to other samples I have seen.

5. Why did you chose to use both BRIEF and SENA , I assume that the FASD children also have been tested with WISC/WAIS? 

6 In my experiance, up to 2/3 of the children with FASD have a ADHD diagnoses i addition, and some of them receiving stimulants. Can`t find any information about that.

7. In line 324 you are using the abbreviation ASD (autism specter disorder), is that correct? 
